# Relationship between visual function and cognitive function in the elderly: A cross-sectional observational study

**Minako Kaido**[1,2]*, **Masaki Fukui**[1,3], **Motoko Kawashima**[1], **Kazuno Negishi**[1], **Kazuo Tsubota**[1,4]

**1** Department of Ophthalmology, Keio University School of Medicine, Tokyo, Japan, **2** Wada Eye Clinic, Chiba, Japan, **3** Tokyo Medical Center, Tokyo, Japan, **4** Tsubota Laboratory, Inc., Tokyo, Japan

* tomoulton777@ff.em-net.ne.jp

**Data Availability Statement:** All relevant data are within the manuscript and its Supporting Information files.

**Funding:** M. Kaido and K. Tsubota hold the patent rights for the method and the apparatus used for

## Abstract

It has been suggested that functional visual acuity (VA) testing may be able to measure both the visual performance and cognitive ability needed for driving and help to reduce the number of road traffic accidents. The aim of this study was to investigate the relationship between visual ability and cognitive function in healthy elderly subjects. The study included 34 eyes with a decimal best-corrected visual acuity (VA) ≥1.0 in 34 subjects (16 men, 18 women; mean age 72.7 ± 6.1 [range, 61–83] years) with the same type of monofocal intraocular lens implant. Using the score on the Japanese version of the Mini-Mental State Examination (MMSE) questionnaire, the subjects were divided into a mild cognitive impairment (MCI) group (score <28) and a normal cognition (NC) group (score ≥28). Visual ability was evaluated by functional VA testing. Functional VA was significantly lower in the MCI group (n = 10) than in the NC group (n = 24; *P*<0.02). There was no significant difference in best-corrected VA between the two groups. High correlations were found between the MMSE score and the logMAR functional VA (r = -0.36, *P* = 0.04), standard deviation of functional VA (r = -0.39, *P* = 0.02), and the visual maintenance ratio (r = 0.34, *P* = 0.048). In summary, despite a good best-corrected VA, deterioration in visual ability was detected in elderly individuals with MCI when measured by the functional VA test. Functional VA could be used to evaluate the integrated visual ability associated with age-related cognitive decline and have applications that help to reduce the disproportionately high rate of road traffic accidents in the elderly.

## Introduction

The population is aging in developed countries on a global scale. Japan is experiencing rapid growth of its aged population, with the elderly (aged over 65 years) comprising 28% of the population according to Japanese Ministry of Health, Labour and Welfare data for 2019 [1]. Therefore, one in four individuals in Japan are elderly, and the normal aging process in this population inevitably causes functional decline in physical ability and memory, potentially hindering the ability to perform everyday tasks.

the measurement of functional visual acuity (US patent no: 255 7470026). Outside the submitted work, Kazuo Tsubota reports he is CEO of Tsubota Laboratory, Inc., Tokyo, Japan. Other authors have no affiliation with any corporation. The funder provided support in the form of salaries for author [KT] but did not have any additional role in the study design, data collection and analysis, decision to publish, or preparation of the manuscript. The specific roles of these authors are articulated in the Author contributions section.

**Competing interests:** We advise that two of the study investigators, Kazuo Tsubota and Minako Kaido, hold the patent rights for the methodology and apparatus used for measurement of functional visual acuity (US Patent Number 7470026). This does need not affect our adherence to PLoS ONE policies on sharing data and materials.

Road traffic accidents (RTAs) caused by elderly drivers have becoming increasingly common in recent years and now pose a serious problem in Japan. Driving a vehicle requires a series of advanced tasks that integrate and process information rapidly and relies on good cognitive skills, including attention, memory, and judgment. Many studies have demonstrated a positive relationship between decline of cognitive function and risk of an RTA [2–7]. Therefore, individuals over the age of 75 years are now required to pass a cognitive function test and attend training for seniors in addition to passing a vision test when renewing their driving licenses in Japan. A diagnosis of dementia results in automatic cancellation of a driving license. Nevertheless, data held by the National Police Agency in Japan show that the rate of fatal RTAs caused by drivers over 75 years of age is more than 2.5 times higher than that for drivers under this age, despite a recent overall decrease in fatal RTAs [8]. Driving requires judgment and operational skills in response to visual information. In general, visual ability and cognitive function are assessed separately; however, a more integrated assessment of these functions may be necessary for driving. Therefore, an effective method is needed to detect changes in integrative function due to subtle age-related cognitive changes and a decline in visual ability specific for driving.

Functional visual acuity (VA) is measured using a test that takes the time axis into account by including use of a joystick and is reportedly useful for assessment of visual function related to daily activities [9–12]. When performing this test, the subject rests the chin on a chin-pad and looks into the testing device, gazes at a target that is automatically displayed for a set period of 2 seconds on a built-in screen, and responds to the target by manipulating the joystick over a set period of 60 seconds. Functional VA testing may be able to measure visual performance in a way that includes the cognitive ability necessary for driving competency. Negishi et al. investigated the relationship between functional VA and the visual field and suggested that functional VA measurement may be applicable to visual function during driving [13]. Similarly, Hiraoka et al. assessed functional VA under mesopic conditions in healthy subjects and suggested that public education on mesopic functional VA may help to reduce the number of RTAs [14]. In this study, we investigated whether or not functional VA measurement can reflect cognitive function in healthy elderly persons.

## Materials and methods

### Subjects

The study included 34 eyes with a decimal best-corrected distance VA ≥1.0 and the same type of monofocal intraocular lens implant (NX-70, Santen Pharmaceutical Co., Ltd, Osaka, Japan) in 34 patients attending the National Hospital Organization Tokyo Medical Center in Tokyo. The patients comprised 16 men and 18 women of mean age 72.7 ± 6.1 (range, 61–83) years. When both eyes were affected, the right eye was studied. Subjects were excluded if they had a history of ophthalmic disease affecting vision, such as definite dry eye based on the 2006 Japanese diagnostic criteria, glaucoma, retinal disease, ocular trauma, or intracranial disease, as were those with a physical disorder that interfered with the skills necessary to manipulate the functional VA test equipment.

This cross-sectional study was approved prospectively by the Institutional Review Board of National Hospital Organization Tokyo Medical Center, Tokyo, Japan, and was performed in accordance with the tenets of the Declaration of Helsinki. Written informed consent was obtained from all subjects after they had received an explanation of the nature of the study and its possible consequences.

## Cognitive function

Cognitive function was measured using the Japanese version of the Mini-Mental State Examination (MMSE; Nihon Bunka Kagakusha, Tokyo, Japan) questionnaire [15]. The MMSE consists of 11 items examining the following functions: orientation to time (Q1, 5 points) and place (Q2, 5 points), registration (Q3, 3 points), attention and calculation (Q4, 5 points), recall (Q5, 3 points), language (Q6, 2 points; Q7, 1 points; Q8: 3 points; Q9, 1 point, Q10, 1 point), and copying (Q11, 1 point). The possible overall score ranges from 0 to 30 points, with a lower score indicating more severe cognitive impairment. A score ≤23 indicates a possibility of dementia (sensitivity 81%, specificity 89%) [16,17], and a score of 24–27 indicates a possibility of mild cognitive impairment (MCI; sensitivity 45%–60%, specificity 65%–90%) [18,19].

## Functional VA measurement system

Visual ability was assessed by measuring dynamic changes in VA continuously for 60 seconds under natural blinking conditions using a functional VA measurement system (AS-28; Kowa, Aichi, Japan; Fig 1A). When performing this test, subjects wear spectacles containing the refractive correction needed to obtain best-corrected VA and delineate an automatically presented Landolt ring orientation on the built-in screen by looking into the machine and manipulating a joystick. The size of the optotype changes by one step depending on the subject's response (Fig 1B). If the response is correct, a smaller optotype is presented as the next stimulus; if the response is incorrect, a larger optotype is presented. The results are recorded as a line graph made up of points joining only the correct answers (Fig 1C).

The outcome parameters in this study included starting VA, functional VA, visual maintenance ratio, standard deviation (SD) of functional VA, response time, and blink frequency [9]. Starting VA is defined as the best-corrected VA measured by the functional VA measurement system. Functional VA is defined as the average of all VA values measured over time. The visual maintenance ratio is defined as the functional VA divided by the starting VA [20]. The SD of the functional VA describes the fluctuation in VA measurements over time. The blink frequency is automatically recorded using the functional VA measurement device. Subjects respond to the target by manipulating a joystick.

## Measurement of wavefront aberrations

Higher-order aberrations were measured using the Hartmann-Shack wavefront aberrometer (Topcon Corp., Tokyo, Japan) to evaluate each subject's optical quality. The higher-order aberration data were analyzed quantitatively in the central 6-mm diameter up to the fourth order by expanding the set of Zernike polynomials. From the Zernike coefficients, the root mean square was calculated to represent the wavefront aberrations. S3 and S4 are the root mean square of the third-order and fourth-order Zernike coefficients, respectively. Total higher-order (S3 + S4) aberration data were recorded.

## Dry eye examination

The Schirmer test was performed without topical anesthesia using a sterilized Schirmer strip (Whatman No. 41, Ayumi Pharmaceutical Corporation, Tokyo, Japan) after completion of all other examinations to confirm the presence or absence of dry eye, which may affect visual function.

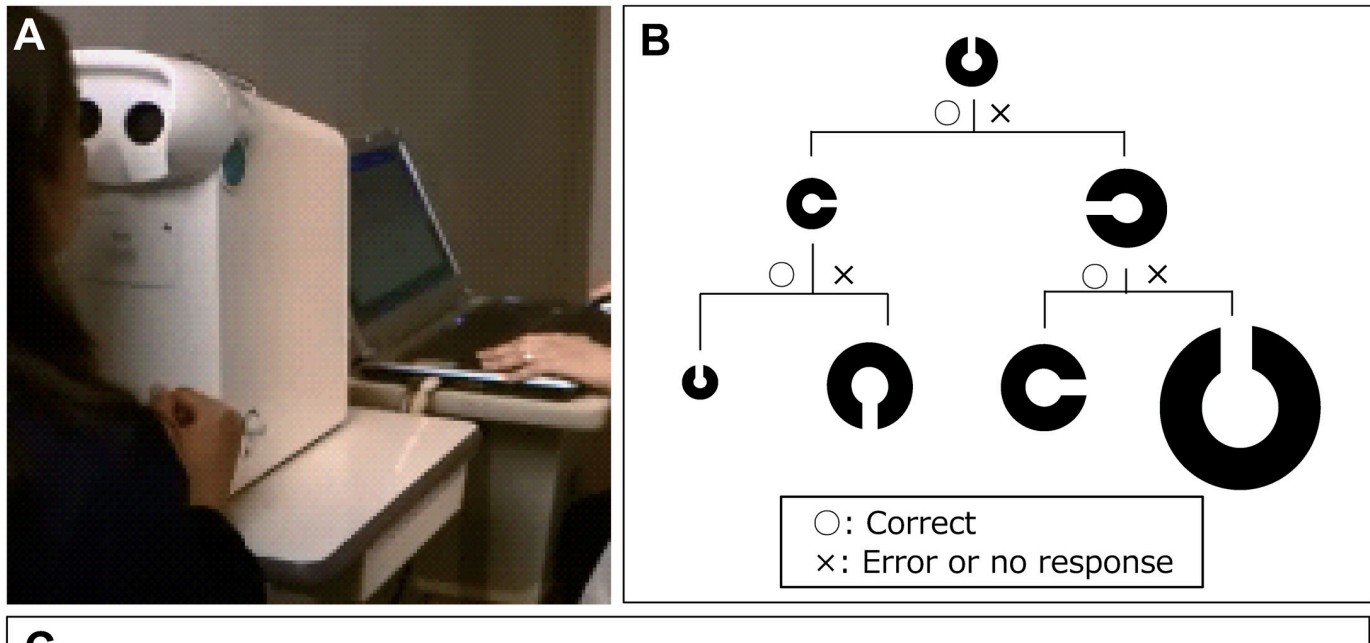

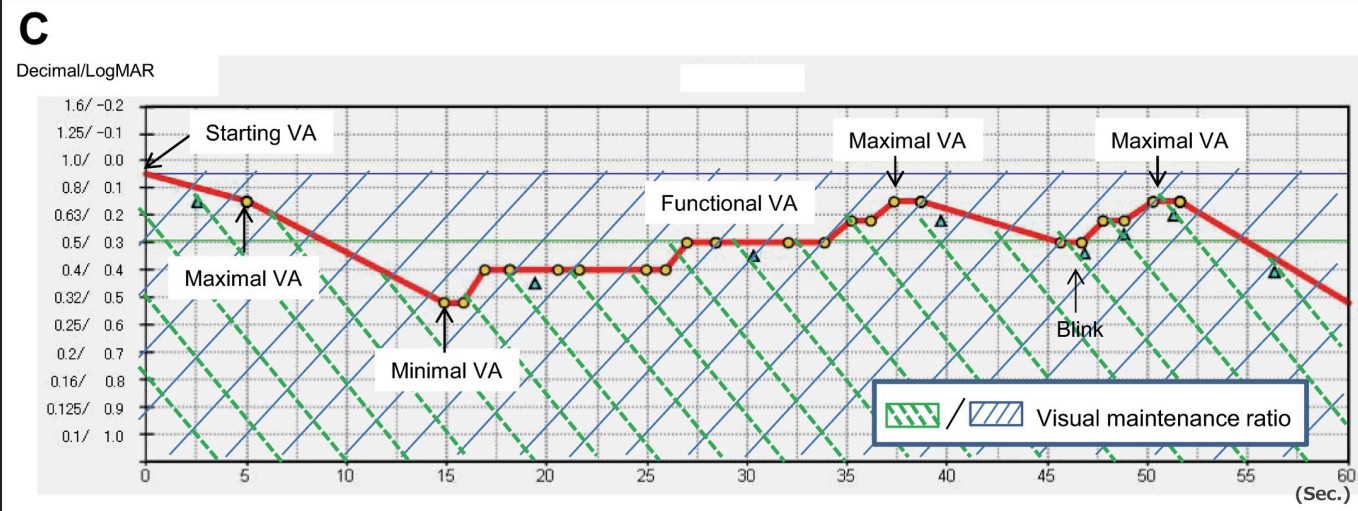

**Fig 1. Functional visual acuity measurement system.** (A) Equipment (AS-28) used to measure functional visual acuity. (B) Optotype display on the functional visual acuity system. (C) The results presented in graphical form. VA, visual acuity; logMAR, logarithm of the minimum angle of resolution.

## Statistical analysis

Subjects with an MMSE score <28 were allocated to an MCI group and those with an MMSE score ≥28 to a normal cognition (NC) group. Characteristics of cognitive functions, such as orientation to time and place, registration, attention and calculation, recall, language, and copying, were compared between the two groups.

The functional VA parameters, total higher-order aberrations, and Schirmer values were compared between the MCI and NC groups. The Student's *t*-test was used for between-group comparisons after confirming that the data were equally distributed using the F-test. The relationship between the functional VA parameters and MMSE scores in the study population overall was examined by Pearson's correlation analysis. A further test of the significance of the correlation coefficient was performed to confirm whether or not the null hypothesis would be

**Table 1. Functional VA parameters in the MCI and NC groups.**

| | MCI group n = 10 | NC group n = 24 | *P*-value |
|---|---|---|---|
| Age (years) | 73.4 ± 7.0 | 72.5 ± 5.8 | 0.69 |
| BCVA at examination (logMAR) | -0.06 ± 0.03 | -0.07 ± 0.03 | 0.41 |
| Preoperative VA (logMAR) | 0.27 ± 0.30 | 0.13 ± 0.20 | 0.11 |
| Schirmer value (mm) | 8.6 ± 5.0 | 11.4 ± 6.6 | 0.24 |
| Higher-order aberration (6 mm; μm) | 0.92 ± 0.28 | 0.78 ± 0.22 | 0.13 |

BCVA, best-corrected visual acuity; logMAR, logarithm of the minimum angle of resolution; MCI, mild cognitive impairment; NC, normal cognition

rejected. SPSS software (version 17.0J for Windows; IBM Corp., Armonk, NY, USA) was used for the statistical analysis. A *P*-value of <0.05 was considered statistically significant.

## Results

### Demographic and ophthalmic characteristics of subjects

Table 1 shows the profile of the subjects in each study group. There were no significant differences in age, best-corrected VA at examination, preoperative VA, Schirmer value, or total higher-order aberrations.

### Cognitive function

The overall distribution of the MMSE scores ranged from 18 to 30. Ten subjects had an MMSE score of <28 (the MCI group) and 24 subjects had a score of ≥28 (the NC group).

Scores for Q1, Q2, Q4, Q5, and Q10 on the MMSE were significantly low in the MCI group but were in the normal range for Q3, Q6, Q7, Q8, Q9, and Q11 in both the MCI and NC groups. Fig 2 shows the distributions of scores for orientation to time and place (sum of Q1 and Q2), registration (Q3), attention and calculation (Q4), recall (Q5), language (sum of Q6, Q7, Q8, Q9, and Q10), and copying (Q11). There were significant differences in orientation to time and place, attention and calculation, and recall between the groups ($P < 0.05$).

### Functional visual acuity

Fig 3 shows the distribution of the functional VA parameters in the MCI and NC groups. Although there was no significant between-group difference in the mean starting VA value (0.05 ± 0.24 vs 0.01 ± 0.08; $P>0.05$), the mean functional VA value was significantly lower and the SD of the functional VA value tended to be lower in the MCI group than in the NC group (0.23 ± 0.13 vs 0.09 ± 0.13; $P = 0.007$ and 0.06 ± 0.04 vs 0.04 ± 0.02; $P = 0.05$, respectively). There was no significant difference in the visual maintenance ratio (0.94 ± 0.06 vs 0.97 ± 0.05; $P = 0.11$), reaction time (1.27 ± 0.18 vs 1.35 ± 0.20; $P = 0.34$), or blink frequency (5.3 ± 3.4 vs 4.4 ± 5.5; $P = 0.64$) between the groups. Fig 4 shows representative cases with MCI and NC.

### Correlation between cognitive function and visual function

No correlation was found between the logMAR (logarithm of the minimum angle of resolution) VA and MMSE score ($P>0.05$; Fig 5A). However, a strong correlations were found between the MMSE score and the logMAR functional VA (r = -0.36, $P = 0.04$; Fig 2B), visual maintenance ratio (r = 0.34, $P = 0.048$; Fig 2C), and SD of the logMAR functional VA (r = -0.39, $P = 0.02$; Fig 2D). Further tests of significance for the correlation coefficient, test statistics, and P-values were 2.195 and 0.036 for MMSE scores and functional VAs, 2.052 and

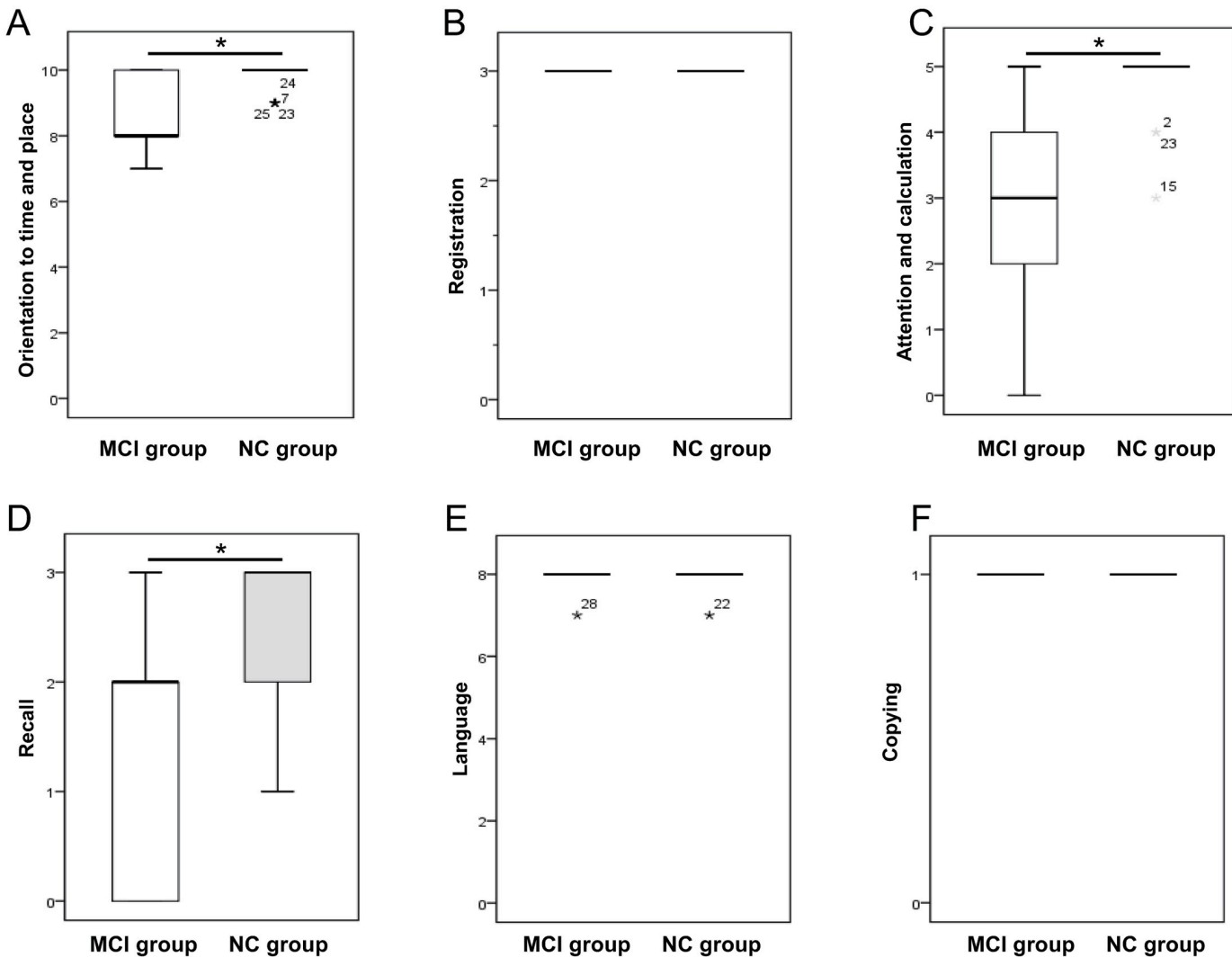

**Fig 2. Distribution of MMSE scores for orientation to time and place, registration, attention and calculation, recall, language, and copying in the MCI and NC groups.** Distribution of orientation to time and place (A), registration (B), attention and calculation (C), recall (D), language (E), and copying (F). MCI, mild cognitive impairment; MMSE, Mini-Mental State Examination; NC, normal cognition.

0.0485 for MMSE scores and the visual maintenance ratio, and 2.089 and 0.0448 for MMSE scores and the SD of functional VAs, respectively, demonstrating that the null hypothesis was rejected. There were no correlations between blink frequency, reaction time, and the MMSE score ($P > 0.05$).

## Discussion

The optical system and central nervous system participate together in visual ability. When considering visual ability in the elderly, it is difficult to evaluate visual function by distinguishing completely between these two systems, given that the elderly have a high incidence of comorbid ocular disease and cognitive impairment. The relationship between visual and cognitive function is well documented in the literature [21–24]. Long-term deterioration of optical quality as a result of cataract [25], open-angle glaucoma [26,27], or age-related macular

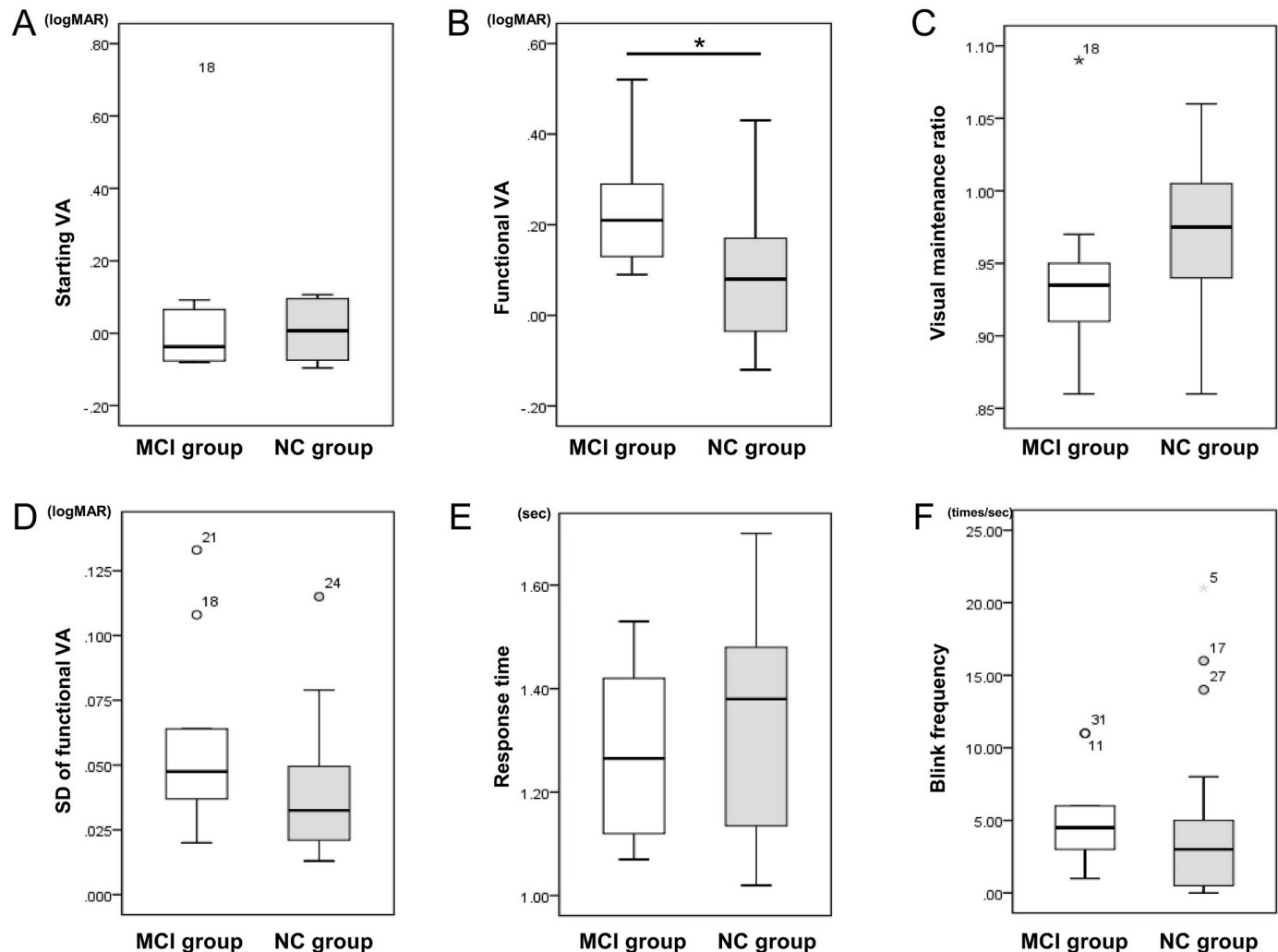

**Fig 3. Distributions of functional visual acuity parameters in the MCI and NC groups.** Distribution of starting visual acuity (A), functional visual acuity (B), visual maintenance ratio (C), SD of functional visual acuity (D), response time (E), and blink frequency (F). logMAR, logarithm of the minimum angle of resolution; MCI, mild cognitive impairment; NC, normal cognition; SD, standard deviation; VA, visual acuity.

degeneration [28,29] decreases brain function, resulting in cognitive impairment. It is also known that vision deteriorates in the absence of organic abnormality in patients with visuospatial agnosia and that some individuals with dementia have symptoms of image degradation due to visuo-constructional and visuo-perceptual dysfunction despite having no ophthalmic abnormality. A morphological stimulus is projected as an indistinct image on the retina, transmitted to the optic nerve, and through two separate pathways in the temporal and parietal association areas to the prefrontal cortex [30]. The original image can be recognized by integrating information and memory from other sensory organs at higher centers and correcting the indistinct image [31]. In this study, we tried to eliminate these optical effects as much as possible by recruiting healthy elderly subjects without any ocular disease and a decimal VA ≥1.0 after cataract surgery and examining the relationship between the changes in cognitive function and visual function associated with physiological aging. The higher-order aberration data confirmed that there was no difference in optical quality between the MCI and NC

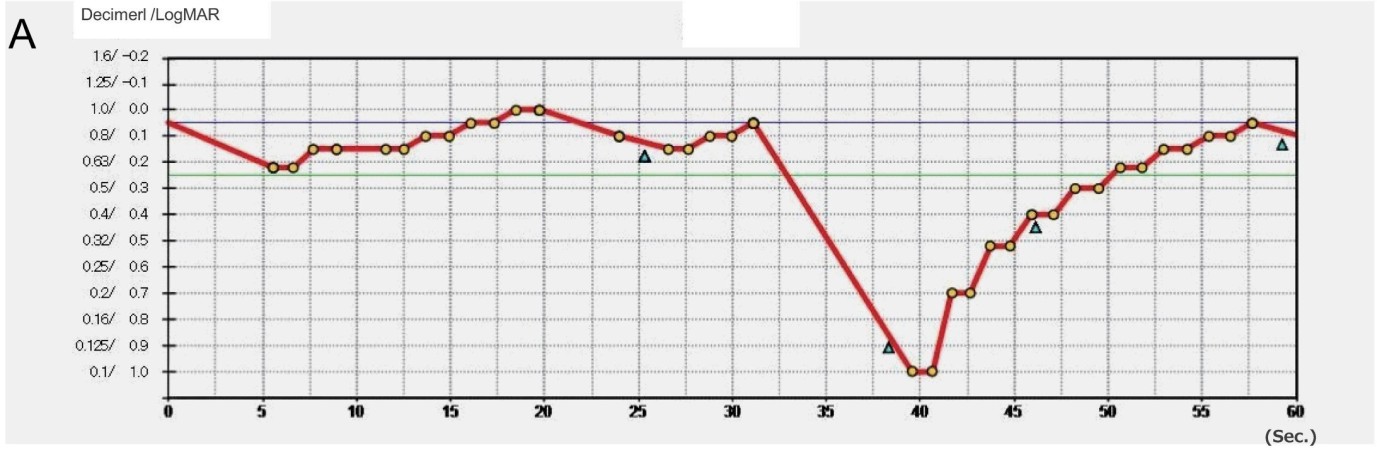

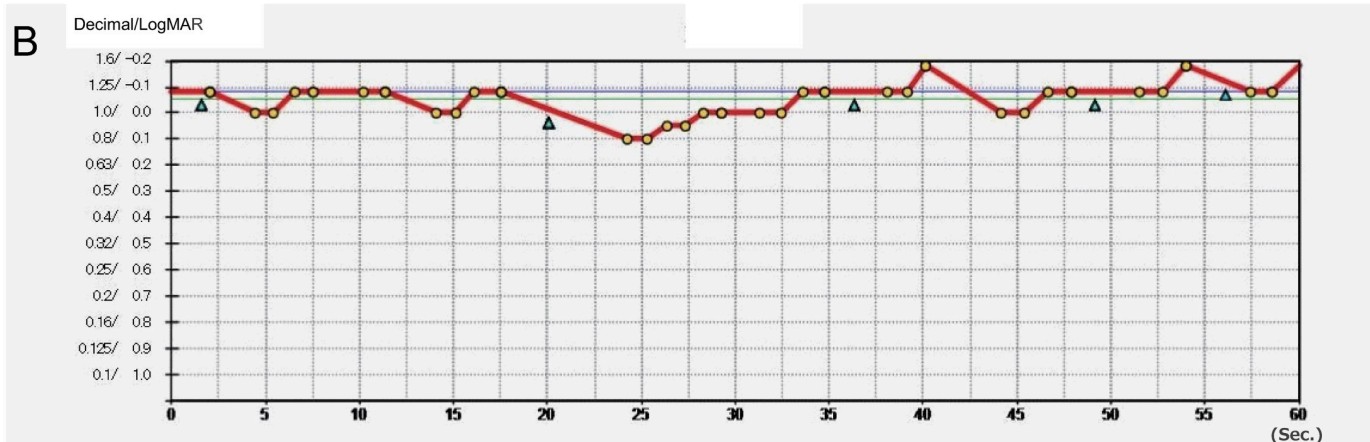

**Fig 4. Functional visual acuity in representative cases with MCI and NC.** (A) A 75-year-old man with an MMSE score of 30. (B) A 71-year-old man with an MMSE score of 18. MCI, mild cognitive impairment; MMSE, Mini-Mental State Examination; NC, normal cognition.

groups, which meant that we could exclude the optical factor and focus on evaluating the impact of the nervous system on visual function.

MCI is an intermediate stage between normal age-related cognitive changes and dementia, in which affected individuals may experience subtle changes in the complex physical and cognitive activities needed to function in society [32,33]. Ten of the 34 healthy elderly subjects in this study were found to have MCI, which suggests that approximately 30% of the elderly population may be unaware that they have early dementia. In our MCI group, the MMSE scores for the domains of orientation, attention and calculation, and recall were significantly decreased while language-related cognitive function was preserved. It is known that cognitive and executive functions, such as working memory, attention, and decision-making, are mediated by the prefrontal cortex [34]. Furthermore, the prefrontal cortex is known to be one of the last cortical regions to mature both anatomically and functionally [34,35] and to be one of the first to deteriorate with normal aging [34,36]. We suspect that it may be hard to recognize individuals with mild dementia given that language-related cognitive function is relatively well preserved in MCI. Furthermore, the cognitive and executive functions performed by the

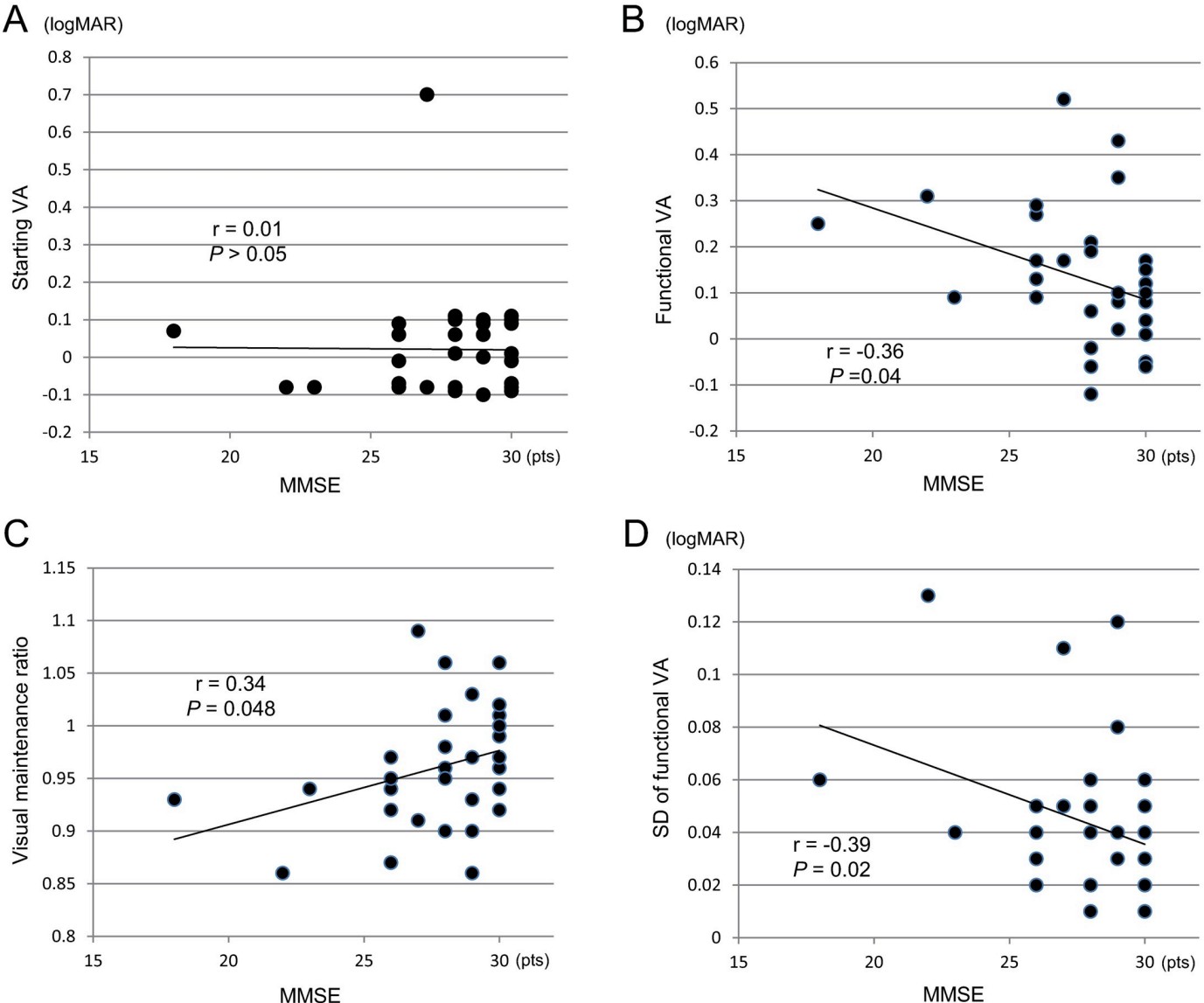

**Fig 5. Correlation between cognitive function and visual function.** (A) Correlation between the logMAR starting VA and the MMSE score. (B) Correlation between the logMAR functional VA and the MMSE score. (C) Correlation between the visual maintenance ratio and the MMSE score. (D) Correlation between the logMAR standard deviation of functional VA and the MMSE score. logMAR, logarithm of the minimum angle of resolution; MMSE, Mini-Mental State Examination; VA, visual acuity.

prefrontal cortex, including recognizing the situation one is in, retaining information temporally, making a decision, and acting upon it, are susceptible to deterioration at an early stage during normal aging. Therefore, integrated function is likely to be reduced in the elderly, and it is important to assess a series of complex functions rather than a single function in this age group.

In our study, the mean functional VA value was significantly lower in the MCI group in the absence of a decrease in best-corrected VA. Moreover, strong correlations were observed between the MMSE score and functional VA, the visual maintenance ratio, and the SD of functional VA. We can infer that the decreased visual performance in the MCI group was caused

by a decline in brain-related function rather than an optical factor, given that the optical quality was the same in both the MCI and NC groups. There have been reports of subclinical visual impairment in visually asymptomatic patients using high sensitivity methods such as contrast sensitivity testing [37–39] that are consistent with our findings. Wieder et al. showed that cognitive function not only correlated with low contrast sensitivity but also with information processing speed and memory among the cognitive domains affecting visual performance [39]. Our finding of decreased MMSE scores for orientation, attention and calculation, and recall in the MCI group suggests that degradation of the prefrontal cortex should be investigated further as a cause of decreased visual ability. During the functional VA test, the subject must intensively and continuously look at the displayed target, where concentration requires "attention and concentration", remembering the target requires "temporary memory", and moving the joystick to the target requires "execution and working memory". This means that the functional VA test can assess integrated function in response to visual stimuli. Driving requires judgment and operational skills with immediate transmission of information, integration and processing of information, and a capacity to move from stimulation to execution. Therefore, age-related deterioration of the prefrontal cortex can increase the risk of an RTA. We propose that the functional VA measurement system could be used as an advanced vision test to assess visual performance in elderly drivers and to detect an intermediate stage between normal age-related cognitive changes and dementia caused by deterioration of the prefrontal cortex.

Unexpectedly, reaction time did not show any correlation with the MMSE score in our study. Considering that degradation of the prefrontal cortex causes impairment in the sequence of perception, cognition, and execution, we expected to find an association between a lower MMSE score and slower reaction time. The prefrontal cortex plays an important role in suppressing unnecessary and inappropriate reactions and focusing on the appropriate reaction when needed [34,40]. Therefore, we infer that the subjects with lower MMSE scores delineated the presented optotype without confirming whether or not the response was correct, given that none of the enrolled subjects had a motor abnormality of the fingers that would interfere with the ability to manipulate the joystick used in the functional VA test. This problem would be exacerbated if there was no delay in reaction time and potentially increase the risk of an RTA.

We used the MMSE to evaluate cognitive function in this study because it is simple to administer and is widely used as a general screening test of cognitive function. However, the emphasis of this test is on memory and language. In view of our finding of deterioration in functional VA parameters in subjects with MCI, which indicate a decline of cognitive functions that are specific to the prefrontal cortex, a further study is needed to investigate the relationship between the domains specific to the prefrontal cortex and practical vision in more depth.

Our study has a few limitations. First, the number of subjects included was small. Moreover, the subjects were not limited to drivers. Second, we focused solely on measurement of cognitive function and visual ability and did not assess the relationship between actual driving performance and cognitive and visual function. However, many other external factors influence the risk of an RTA, such as driving performance [41,42], environmental road conditions [43,44], and traffic conditions [45,46]. Therefore, more research is needed in a large number of subjects to investigate the relationship between MCI and risk of an RTA.

In conclusion, approximately 30% of healthy elderly subjects in this study were found to have MCI. Subjects with suspected MCI were found to have decreased cognitive functions relating to orientation, attention and calculation, and recall despite intact language-related cognitive function. These subjects also showed deterioration of visual ability when measured by the functional VA test despite having no decrease in best-corrected VA. Measurement of

functional VA may be useful for assessment of the integrated visual ability associated with cognitive function.

## Supporting information

**S1 Data.**
(XLSX)

## Acknowledgments

The authors thank Dr. Yoshinobu Mizuno, Department of Ophthalmology, Teikyo University School of Medicine, Tokyo, Japan for advice on statistical analysis.

## Author Contributions

**Conceptualization:** Minako Kaido.

**Data curation:** Masaki Fukui.

**Formal analysis:** Minako Kaido.

**Funding acquisition:** Masaki Fukui.

**Investigation:** Minako Kaido.

**Methodology:** Minako Kaido, Masaki Fukui, Kazuo Tsubota.

**Project administration:** Minako Kaido, Masaki Fukui.

**Resources:** Minako Kaido, Masaki Fukui.

**Software:** Minako Kaido, Masaki Fukui.

**Supervision:** Minako Kaido, Motoko Kawashima, Kazuno Negishi, Kazuo Tsubota.

**Validation:** Minako Kaido.

**Visualization:** Minako Kaido.

**Writing – original draft:** Minako Kaido.

**Writing – review & editing:** Masaki Fukui, Motoko Kawashima, Kazuno Negishi, Kazuo Tsubota.

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
