## [Decision Letter · Decision Letter 0]

4 Mar 2020

PONE-D-20-03098

Relationship between visual function and cognitive function in the elderly: a cross-sectional observational study

PLOS ONE

Dear Dr Kaido,

Thank you for submitting your manuscript to PLOS ONE. After careful consideration, we feel that it has merit but does not fully meet PLOS ONE’s publication criteria as it currently stands. Therefore, we invite you to submit a revised version of the manuscript that addresses the points raised during the review process.

We would appreciate receiving your revised manuscript by Apr 18 2020 11:59PM. To enhance the reproducibility of your results, we recommend that if applicable you deposit your laboratory protocols in protocols.io, where a protocol can be assigned its own identifier (DOI) such that it can be cited independently in the future. For instructions see: http://journals.plos.org/plosone/s/submission-guidelines#loc-laboratory-protocols

We look forward to receiving your revised manuscript.

Kind regards,

Feng Chen

Academic Editor

PLOS ONE

Journal Requirements:

3. We note that you have a patent relating to material pertinent to this article. Please provide an amended statement of Competing Interests to declare this patent (with details including name and number), along with any other relevant declarations relating to employment, consultancy, patents, products in development or modified products etc. Please confirm that this does not alter your adherence to all PLOS ONE policies on sharing data and materials, as detailed online in our guide for authors http://journals.plos.org/plosone/s/competing-interests by including the following statement: "This does not alter our adherence to  PLOS ONE policies on sharing data and materials.” If there are restrictions on sharing of data and/or materials, please state these. Please note that we cannot proceed with consideration of your article until this information has been declared.

4. Thank you for stating the following in the Financial Disclosure section:

"M. Kaido and K. Tsubota hold the patent rights for the method and the apparatus used for the measurement of functional visual acuity (US patent no: 255 7470026). Outside the submitted work, Kazuo Tsubota reports he is CEO of Tsubota Laboratory, Inc., Tokyo, Japan.

Other authors have no affiliation with any corporation."

We note that one or more of the authors are employed by a commercial company: Tsubota Laboratory, Inc.

Reviewers' comments:

Reviewer's Responses to Questions

**Comments to the Author**

1. Is the manuscript technically sound, and do the data support the conclusions?

Reviewer #1: Yes

Reviewer #2: Partly

2. Has the statistical analysis been performed appropriately and rigorously? 

Reviewer #1: No

Reviewer #2: I Don't Know

3. Have the authors made all data underlying the findings in their manuscript fully available?

Reviewer #1: Yes

Reviewer #2: Yes

4. Is the manuscript presented in an intelligible fashion and written in standard English?

Reviewer #1: Yes

Reviewer #2: Yes

5. Review Comments to the Author

Reviewer #1: The topic of this study is refreshing to focus on the relationship between visual ability and cognitive function. The results reported are still somewhat exploratory and needed to be examined further due to the following aspects:

1) The allocation of subjects is a bit confusing. In Cognitive Function, “a score<=23 indicates a possibility of dementia…, and a score of 24-28 indicates a possibility of MCI”. However, in discussion section, subjects with a MMSE sore<28 were allocated to a MCI group. How did the author consider here? Or was it not rigorous enough.

2) When the first paragraph of “Discussion” was read, this reviewer understood that the higher-order aberration was used to exclude the optical factor. It would be clearer that the author explained previously in “Measurement of wavefront aberrations”. It is regrettable that this reviewer still do not understand the role of Schirmer value.

3) The author focused on the measurement of cognitive function and visual ability. But there are many other external factors affecting them, such as driving performance, road environment, traffic condition...More researches are needed. The finding was expected to be used to reduce road traffic rate and assess visual performance in elderly drivers. Most analyses focused on explaining the relationship from a medical perspective, not from a perspective of accident prevention. And 34 subjected selected were unclear whether they were drivers.

4）Detailed the experiment section, such as providing the necessary figures to show the experiment and equipment, explaining the experiment procedure.

5) Statistical analysis used in the article was simple. The results shown in figure 2, R or R2？Anyhow, all the values are not good enough, maybe which is due to the small data samples. Please provide more explanation or comparison to previous studies. The figures need to be redrawn, such as adding axis lables.

6) The conclusion section is missing, maybe the current analysis section could be divided into two sections.

Reviewer #2: The topic of this paper is interesting and the methods sound. The results are useful and meaningful. There are several suggestions to improve this paper.

1. Some references are needed for student’s test. And the authors need to mention that why student’s test is used instead of ANOVA. The authors could refer to the following ones.

[1] Examining the safety of trucks under crosswind at bridge-tunnel section: A driving simulator study, Tunnelling and Underground Space Technology, 2019, 92, 103034. https://doi.org/10.1016/j.tust.2019.103034

[2] Examining the influence of decorated sidewaall in road tunnels using fMRI technology, Tunnelling and Underground Space Technology, Volume 99, 2020, https://doi.org/10.1016/j.tust.2020.103362

2. For Fig.1, Box-plot with outlier maybe is preferred.

6. PLOS authors have the option to publish the peer review history of their article (what does this mean?). If published, this will include your full peer review and any attached files.

Reviewer #1: No

Reviewer #2: No

---

## [Author Response · Author response to Decision Letter 0]

14 Apr 2020

Reviewer #1: The topic of this study is refreshing to focus on the relationship between visual ability and cognitive function. The results reported are still somewhat exploratory and needed to be examined further due to the following aspects:

1) The allocation of subjects is a bit confusing. In Cognitive Function, “a score<=23 indicates a possibility of dementia…, and a score of 24-28 indicates a possibility of MCI”. However, in discussion section, subjects with a MMSE sore<28 were allocated to a MCI group. How did the author consider here? Or was it not rigorous enough.

Thank you for pointing it out. It was just an unintended mistake. Thus, we revised it as follows: “…a score of 24–27 indicates a possibility of mild cognitive impairment...”

2) When the first paragraph of “Discussion” was read, this reviewer understood that the higher-order aberration was used to exclude the optical factor. It would be clearer that the author explained previously in “Measurement of wavefront aberrations”. It is regrettable that this reviewer still do not understand the role of Schirmer value.

Please note that we explained the purpose for the inspection of wavefront aberrations as follows: “Higher-order aberrations were measured using the Hartmann-Shack wavefront aberrometer (Topcon Corp., Tokyo, Japan) to evaluate each subject’s optical quality.”

We also added the explanation of Schirmer test as follows: “The Schirmer test was performed without topical anesthesia using a sterilized Schirmer strip (Whatman No. 41, Ayumi Pharmaceutical Corporation, Tokyo, Japan) after completion of all other examinations to confirm the presence or absence of dry eye, which may affect visual function.”

3) The author focused on the measurement of cognitive function and visual ability. But there are many other external factors affecting them, such as driving performance, road environment, traffic condition...More researches are needed. The finding was expected to be used to reduce road traffic rate and assess visual performance in elderly drivers. Most analyses focused on explaining the relationship from a medical perspective, not from a perspective of accident prevention. And 34 subjected selected were unclear whether they were drivers.

We agree with you. Thus, we added more details in the limitation section as follows: “First, the number of subjects included was small. Moreover, the subjects were not limited to drivers. Second, we focused solely on measurement of cognitive function and visual ability and did not assess the relationship between actual driving performance and cognitive and visual function. However, many other external factors influence the risk of an RTA, such as driving performance [41,42], environmental road conditions [43,44], and traffic conditions [45,46].”

4）Detailed the experiment section, such as providing the necessary figures to show the experiment and equipment, explaining the experiment procedure.

Please note that Fig 1 is provided to show (A) Equipment (AS-28) used to measure functional visual acuity. (B) Optotype display on the functional visual acuity system. (C) The results presented in graphical form. 

We also presented the representative cases with MCI and NC in Figure 4.

5) Statistical analysis used in the article was simple. The results shown in figure 2, R or R2？Anyhow, all the values are not good enough, maybe which is due to the small data samples. Please provide more explanation or comparison to previous studies. The figures need to be redrawn, such as adding axis lables.

Thank you for your constructive comment. The study sample is small for the statistical analysis. Thus, in the analysis of 2-group comparison, we confirmed that data is equally distributed using F-test, and then conducted the analysis of 2-group comparison by using t-test. In the correlation of MMSE scores and functional VA parameters, we conducted a test of no correlation. Correlation coefficient, test statistics and P values were -0.362, 2.195 and 0.036 between MMSE scores and functional VAs, 0.340, 2.052 and 0.0485 between MMSE scores and VMRs, -0.346, 2.089 and 0.0448 between MMSE scores and SD of functional VAs, respectively. Hence, the null hypothesis was rejected, and the data was not uncorrelated.

Please note that we added the additional statistical methods in the statistical section as follows: “The functional VA parameters, total higher-order aberrations, and Schirmer values were compared between the MCI and NC groups. The Student’s t-test was used for between-group comparisons after confirming that the data were equally distributed using the F-test. The relationship between the functional VA parameters and MMSE scores in the study population overall was examined by Pearson’s correlation analysis. A further test of the significance of the correlation coefficient was performed to confirm whether or not the null hypothesis would be rejected.” With the additional information, we added in the result section of “Correlation between cognitive function and visual function” as follows: “Further tests of significance for the correlation coefficient, test statistics, and P-values were 2.195 and 0.036 for MMSE scores and functional VAs, 2.052 and 0.0485 for MMSE scores and the visual maintenance ratio, and 2.089 and 0.0448 for MMSE scores and the SD of functional VAs, respectively, demonstrating that the null hypothesis was rejected.”

We added the new figure of the distribution of the functional VA parameters.

We also cited more references to show the relationship between cognitive and visual functions in the visually asymptomatic subjects as follows: “There have been reports of subclinical visual impairment in visually asymptomatic patients using high sensitivity methods such as contrast sensitivity testing [37-39] that are consistent with our findings. Wieder et al. showed that cognitive function not only correlated with low contrast sensitivity but also with information processing speed and memory among the cognitive domains affecting visual performance [39].” 

We added the axis labels in Fig. 5. 

6) The conclusion section is missing, maybe the current analysis section could be divided into two sections.

Please note that we mentioned in the section of statistical analysis that each function of the MMSE test was compared between the two groups. We also divided into two sections as follows: “Subjects with an MMSE score <28 were allocated to an MCI group and those with an MMSE score ≥28 to a normal cognition (NC) group. Characteristics of cognitive functions, such as orientation to time and place, registration, attention and calculation, recall, language, and copying, were compared between the two groups.

The functional VA parameters, total higher-order aberrations, and Schirmer values were compared between the MCI and NC groups. The Student’s t-test was used for between-group comparisons after confirming that the data were equally distributed using the F-test. The relationship between the functional VA parameters and MMSE scores in the study population overall was examined by Pearson’s correlation analysis. A further test of the significance of the correlation coefficient was performed to confirm whether or not the null hypothesis would be rejected.”

Reviewer #2: The topic of this paper is interesting and the methods sound. The results are useful and meaningful. There are several suggestions to improve this paper.

1. Some references are needed for student’s test. And the authors need to mention that why student’s test is used instead of ANOVA. The authors could refer to the following ones.

[1] Examining the safety of trucks under crosswind at bridge-tunnel section: A driving simulator study, Tunnelling and Underground Space Technology, 2019, 92, 103034. https://doi.org/10.1016/j.tust.2019.103034

[2] Examining the influence of decorated sidewaall in road tunnels using fMRI technology, Tunnelling and Underground Space Technology, Volume 99, 2020, https://doi.org/10.1016/j.tust.2020.103362

Please note that we cited the references, showing the studies of 2-group comparison using t-test as follows: “There have been reports of subclinical visual impairment in visually asymptomatic patients using high sensitivity methods such as contrast sensitivity testing [37-39] that are consistent with our findings. Wieder et al. showed that cognitive function not only correlated with low contrast sensitivity but also with information processing speed and memory among the cognitive domains affecting visual performance [39].”

We also changed the statistical methods and procedures clearly in the section of Statistical analysis as follows: “Subjects with an MMSE score <28 were allocated to an MCI group and those with an MMSE score ≥28 to a normal cognition (NC) group. Characteristics of cognitive functions, such as orientation to time and place, registration, attention and calculation, recall, language, and copying, were compared between the two groups.

The functional VA parameters, total higher-order aberrations, and Schirmer values were compared between the MCI and NC groups. The Student’s t-test was used for between-group comparisons after confirming that the data were equally distributed using the F-test. The relationship between the functional VA parameters and MMSE scores in the study population overall was examined by Pearson’s correlation analysis. A further test of the significance of the correlation coefficient was performed to confirm whether or not the null hypothesis would be rejected.”

We wrote the limitation more clearly as follows: “Our study has a few limitations. First, the number of subjects included was small. Moreover, the subjects were not limited to drivers. Second, we focused solely on measurement of cognitive function and visual ability and did not assess the relationship between actual driving performance and cognitive and visual function. However, many other external factors influence the risk of an RTA, such as driving performance [41,42], environmental road conditions [43,44], and traffic conditions [45,46]. Therefore, more research is needed in a large number of subjects to investigate the relationship between MCI and risk of an RTA.”

We also cited the articles as you recommended.

2. For Fig.1, Box-plot with outlier maybe is preferred.

 Please note that the figure was changed to box-plot with outlier, as recommended.

---

## [Decision Letter · Decision Letter 1]

5 May 2020

Relationship between visual function and cognitive function in the elderly: a cross-sectional observational study

PONE-D-20-03098R1

Dear Dr. Kaido,

We are pleased to inform you that your manuscript has been judged scientifically suitable for publication and will be formally accepted for publication once it complies with all outstanding technical requirements.

With kind regards,

Feng Chen

Academic Editor

PLOS ONE

Additional Editor Comments (optional):

Reviewers' comments:

Reviewer's Responses to Questions

**Comments to the Author**

1. If the authors have adequately addressed your comments raised in a previous round of review and you feel that this manuscript is now acceptable for publication, you may indicate that here to bypass the “Comments to the Author” section, enter your conflict of interest statement in the “Confidential to Editor” section, and submit your "Accept" recommendation.

Reviewer #1: All comments have been addressed

Reviewer #2: All comments have been addressed

2. Is the manuscript technically sound, and do the data support the conclusions?

Reviewer #1: Yes

Reviewer #2: Yes

3. Has the statistical analysis been performed appropriately and rigorously? 

Reviewer #1: Yes

Reviewer #2: Yes

4. Have the authors made all data underlying the findings in their manuscript fully available?

Reviewer #1: Yes

Reviewer #2: Yes

5. Is the manuscript presented in an intelligible fashion and written in standard English?

Reviewer #1: Yes

Reviewer #2: Yes

6. Review Comments to the Author

Reviewer #1: The revised manuscript has addressed review's comments well.

However, I still suggest that the paper need include a separate Conclusion section.

Reviewer #2: (No Response)

7. PLOS authors have the option to publish the peer review history of their article (what does this mean?). If published, this will include your full peer review and any attached files.

Reviewer #1: No

Reviewer #2: No

---

## [Editor Report · Acceptance letter]

7 May 2020

PONE-D-20-03098R1 

Relationship between visual function and cognitive function in the elderly: a cross-sectional observational study 

Dear Dr. Kaido:

I am pleased to inform you that your manuscript has been deemed suitable for publication in PLOS ONE. Congratulations! Your manuscript is now with our production department. 

With kind regards,

on behalf of

Dr. Feng Chen 

Academic Editor

PLOS ONE